# Perspective Studies on Perugino's and Raffaello's Painted Architecture

**Fabio Colonnese** 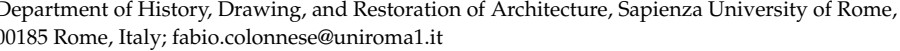

Department of History, Drawing, and Restoration of Architecture, Sapienza University of Rome, 00185 Rome, Italy; fabio.colonnese@uniroma1.it

**Abstract:** The architecture that the Renaissance artists depicted in their works constitutes a vast reservoir of formal solutions that influenced (and were reciprocally influenced by) built architecture. Generally painted according to a rigorous perspective structure, most painted architecture can be restituted and modelled to become part of the Virtual Heritage that develops and extends its knowledge to a wider range of people and scholars. These procedures are here applied to some of the works of Pietro Perugino and Raffaello, a master and his student, in order to define their specific approach to composition, perspective, and architecture. The application of these procedures produced some primary results—the architecture restitutions in plan and elevation or section—and some secondary results concerning the way painted architecture was conceived, represented in perspective, and received as well as it was related to actual architecture.

**Keywords:** perspective; perspective restitution; Pietro Perugino; Raffaello; painted architecture; Virtual Heritage

## 1. Introduction

Despite their basic role in contextualising an event (Lillie 2014), buildings depicted in paintings may have many other different functions, from narrative to allegoric and symbolic (Benelli 2014), for they are often strictly intertwined with the *historia* staged. As Lupi (2017, p. 3) states, "the narrative 'activates' the fictive architecture, helping us to decode its meanings and messages as a setting, while the architecture has the communicative ability to reiterate, clarify and, crucially, intensify the message, thereby strengthening its persuasiveness".

After Giotto's age (Hoffmann 2010), perspective was used to innovate traditional figurative models and to connote painted architecture. For example, Andrea Mantegna's *St. Zeno Altarpiece* (1456–1459) translated the sculptural and three-dimensional model of Donatello's altar in the Basilica of S. Antonio in Padua into a flat perspective representation. Despite structuring it like a traditional medieval polyptic, he interpreted the external architectural frame as the facade of a building and painted the interior in perspective between the supports of the frame (Komatsubara 2014).

From a semantic point of view, in the sacred representations perspective was often used to put in the scene a transcendent space. Florentine tabernacles adopted the perspective to evoke a sort of magical, devotional space (Davies 2013) while after Masaccio's *Trinity* (1426–1428) in S. Maria Novella, Florence (Camerota 2019), some altarpieces were designed as portals or windows (Blum 2008) creating a virtual and illusory expansion of real space: "Often appearing as frames, imitating the liturgical space, fictive architectures introduce a mental and devotional space and act as metaphors of the mind's eye" (Zaru 2018, p. 35).

Thanks to the diffusion of perspective, most of the Renaissance paintings were conceived in a continuous and homogeneous pictorial space, in which the immersed bodies can be represented coherently through the application of projective principles (Mitrovic 2004). In such a scenario, painted architecture is generally based on actual projects, especially when the artists were architects, too. Although aimed at representation and not

construction, many of these projects were consistently designed in three dimensions before being depicted according to a specific point of view. While the original projects are generally lost or systematically destroyed like most of the preparatory works, their image often allows us to reconstruct them by applying the procedure of perspective restitution. Inspired by Guidubaldo del Monte's (1545–1607) studies (Sinisgalli 1978; Andersen 2013), this procedure was developed by Simon Stevin (1548–1620) "to find the eye when a plane figure and its perspective are given" (Struik 1958, vol. II, p. 8) and later scientifically defined by Johann Heinrich Lambert (1728–1777) as the inverse rules of perspective (Rapp 2008). Although initially used for mathematical speculations which only occasionally were applied to painting, it was later addressed to military application to reconstruct the coastline or enemy fortifications from distant viewpoints, eventually developing the modern photogrammetry.

The inverse perspective is based on the application of the linear perspective, in what Leon Battista Alberti called the *costruzione legittima* (Grayson 1964). Briefly, the extension of converging lines representing parallel edges foreshortening in the pictorial space individuates the main Vanishing Point and, consequently, the Horizon Line passing through it; then, by assuming a specific ratio for one of the painted elements, a Lateral Vanishing Point for the 45° horizontal lines can be found on the Horizon Line. Generally, a tile of the chequered floor is assumed as a square and one of its diagonals is traced and extended to the Horizon Line to this scope. The distance between the two Vanishing Points corresponds to the Main Distance, the distance between the Point of View and the Picture Plane, and is adopted as a radius to trace the Circle of Distance around the main Vanishing Point. This allows to place the Point of View in the real space on the perpendicular line coming out of the main Vanishing Point on the picture and to restitute the painted elements in orthogonal projection (Figure 1).

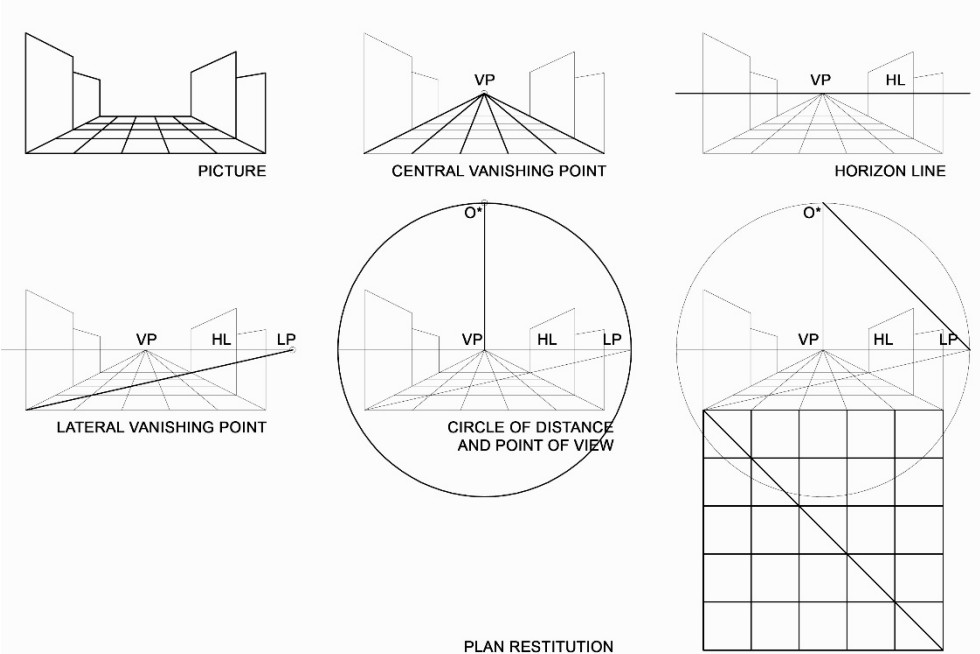

**Figure 1.** Preliminary stages of the procedure of perspective restitution to reconstruct the perspective structure of a picture and draw the plan of its chequered floor.

In the 20th-century, this procedure fuelled an innovative analytical method, and this method was adopted by historians to investigate the form of the famous ideal cities (Krautheimer 1994; Di Bernardino 2015) or the design of the pictorial space, such Wittkover and Carter (1953) with Piero della Francesca's *Flagellation.* Added to this, by providing data on its spatial consistency, the restitution makes it possible to identify the deviations

from the canonical perspective structure, like in Giovan Battista Piranesi's graphic works (Wilton-Ely 1993; Rapp 2008) and helps to attribute a meaning to them, from technical to visual and symbolic levels (Veltman and Keele 1986; Kemp 2005).

The method is also used just to learn more about compositional criteria. For example, some of Vincenzo Fasolo's exercises show the intent "to understand and express, by almost subjectively redesigning it, the organization of the structures of spaces and elements" (Bruschi 2001, pp. 78–79), often over the metric accuracy of and philological rigour. The procedure of perspective restitution allows not only to reconstruct the painted building in plan and elevation (Paris 2000) but also to reveal hidden information otherwise inaccessible and potentially to disseminate the results to a wider audience, according to the operational principles of Virtual Heritage. In addition, as the procedure works as a critical device to question the entire work, it indirectly provides information on the knowledge and modus operandi of the artist, the client's expectations, and the receptive and interpretative skills of the spectators.

Finally, the perspective restitution allows one to interact with the pictorial space of the work in visual terms (White 1958), either as a scientist, by elaborating visual hypotheses consistent with the structure of the work or as an artist, allowing contemporary artists to engage themselves with the cultural heritage of an entire community. In this sense, the applications experimented with by the Spanish architects Flores and Prats (2008) demonstrate the design opportunities hidden in the Dutch interiors depicted in the 17th-century.

On the anniversary of the death of Raffaello (1483–1520), the author was invited to research some of the buildings painted by Pietro Vannucci known as Perugino (1448–1523) and by Raffaello himself, aiming at detecting elements of continuity and innovation both in architecture and perspective. While Perugino's and Raffaello's paintings have been extensively analysed, their architecture has only occasionally been studied (Frommel 1984; Niebaum 2011) and restituted (Aterini et al. 2009), especially concerning the centric plan churches (Portoghesi 1970; Spagnesi et al. 1984).

After this *Introduction*, Perugino's and Raffaello's paintings with canopies and loggias, whose perspective structure is evident, are described and analysed in order to define a homogeneous field of application. Added to these, other Raffaello's works are described to demonstrate his specific typological contribution to the *architectura picta*. Subsequently, the *Perspective analysis and restitution* procedure are briefly described and applied to the chosen works, resulting in digital restitutions; in particular, an attempt to virtually complete an unfinished element of Raffaello's *The Disputa* is used to test its intrinsic perspective consistency; then, some *Considerations* triggered by the application of the procedure involving technical and cultural aspects of the works are discussed. In particular, the comparison between the mathematical structure of the perspective and the elements of the composition provided information on methodological-operational issues and specific aspects of the single works, disclosing the hidden conflict between the medium of perspective and the figurative program. Finally, the *Conclusions* present a summary of this study and its main results.

## 2. *Architectura picta* by Perugino and Raffaello

*The Delivery of the Keys* (1481–1482; Figure 2a) in the Sistine Chapel, Rome, testifies that Perugino is one of the "most brilliant connoisseurs" (Frommel 1984, p. 15) of the revolutionary architecture of Filippo Brunelleschi and Leon Battista Alberti; his skills are shown in the perspective construction, and the interpretative ability to portray Roman antiquities and invent modern architecture. The fresco is divided horizontally into two halves, separated by the low wall that delimits the square at the bottom. The lower half is occupied by the figures of the *historia* in the foreground, while others, further back, introduce secondary episodes or visually measure the depth of pictorial space. The upper half is dedicated to architecture and landscape, with the Umbrian landscape in the distance. In the middle, a five-step base supports an octagonal temple covered by a Florentine

dome and equipped with four baldachins on free columns, while two "restored" Arches of Constantine stand at its sides.

This fresco presents many of the painting topics developed in the previous decades, such as the figurative problem linked to the perspective representation of a building in the middle, where the perspective lines converge, and it is harder to show the depth of bodies (Colonnese and Carpiceci 2022). In this sense, the four baldachins (assuming a fourth one is hidden behind) not only extend the structure in the three-dimensional space but also present the same element from different points of view, such as in an architectural design in double (orthogonal) projection. Nevertheless, Perugino's temple has a primary symbolic and decorative task.

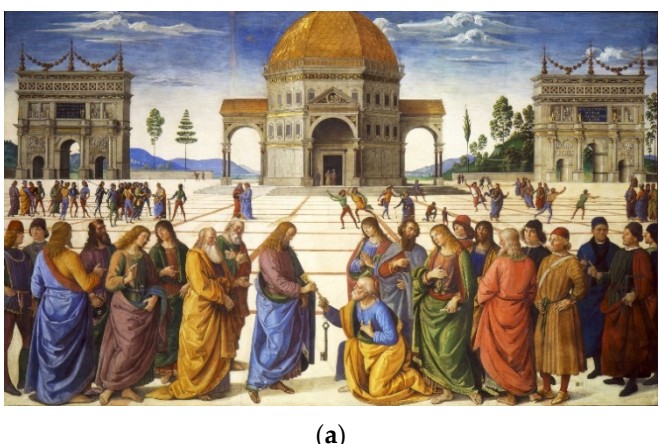 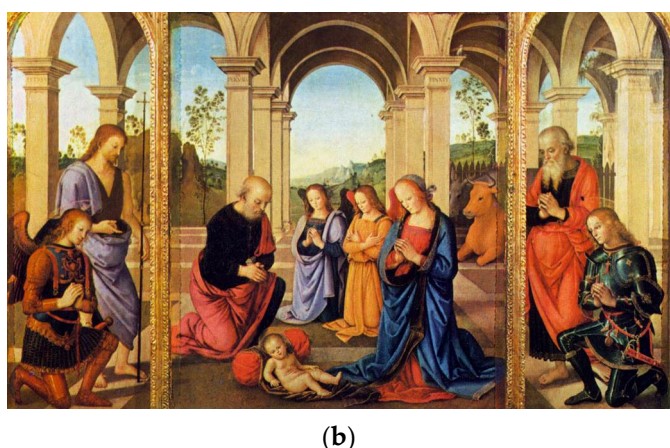

| (**a**) | (**b**) |

**Figure 2.** Pietro Vannucci, known as Perugino, (**a**) The Delivery of the Keys, 1481–82, Sistine Chapel, Vatican City, Rome; (**b**) The Nativity (det.), 1491, Albani-Torlonia polyptych, Villa Albani, Rome.

The fictive condition of Perugino's painted architecture is evident in the two primary architectural devices that he designed and developed in his paintings over the years. The former is a wooden or masonry hut on pillars freestanding on the grass, specifically designed to house "nativities" and "adorations"; the latter is a canopy or loggia-like structure composed of free Tuscan pillars disposed on a grid plan, and supporting cross vaults.

In 15 works at least, Perugino developed the latter type of structure according to a wide range of configurations. It can appear as a simple two-dimensional backdrop in the background; sometimes, it is limited to a single span, such as a *baldacchino*, often around octagonal temples; sometimes, it is multiplied to form a tunnel-like architecture made of six spans (*The Nativity* of the Albani-Torlonia polyptych) or to create a loggia or even a sort of hypostyle hall. In *The Last Supper of Fuligno* (1485–1493) in S. Onofrio, Florence, the structure is combined with an open box, whose top coincides with the perspective horizon that houses Christ and the apostles. In *The Nativity* of the Albani-Torlonia polyptych (1491; Figure 2b), such a modular structure creates a wall, a baldachin, a loggia, and finally, a ruined stable with the vault collapsed.

Early traces of the young Raffaello in Perugino's work can be found in *The Nativity of Mary*, a predella from the Fano altarpiece (1497). As a whole, the panels of the Fano altarpiece show a slight change in the *architectura picta*, which, in apparent contradiction with the previous works, offers a large variety of architectural scenes (generally designed on regular grid plans). *The Annunciation* depicts a gallery open to the landscape behind it, with pillars and pilasters that configure a structure, not unlike the prototype of a Vitruvian atrium. *The Stories of the Virgin* shows a square bedroom with pilasters, a coffered ceiling, and two arches open onto the landscape covered by a barrel vault. *The Presentation* shows the exterior of a temple with a gallery on Ionic columns. *The Marriage* presents the exterior space of an octagonal Ionic temple with four galleries set as a cross in plan (Figure 3). Like frames of a single sequence, these scenes present a horizon line fixed at the eye level of the figures; a broad and descriptive field of view; and figures that share their colours, light,

shadow effects, and plasticity with architecture. A similar approach is found in the late Polyptych of St. Augustine (1502–1523; Figure 4), in the usual cross-shape octagonal church that houses *The Presentation of Jesus in the Temple*.

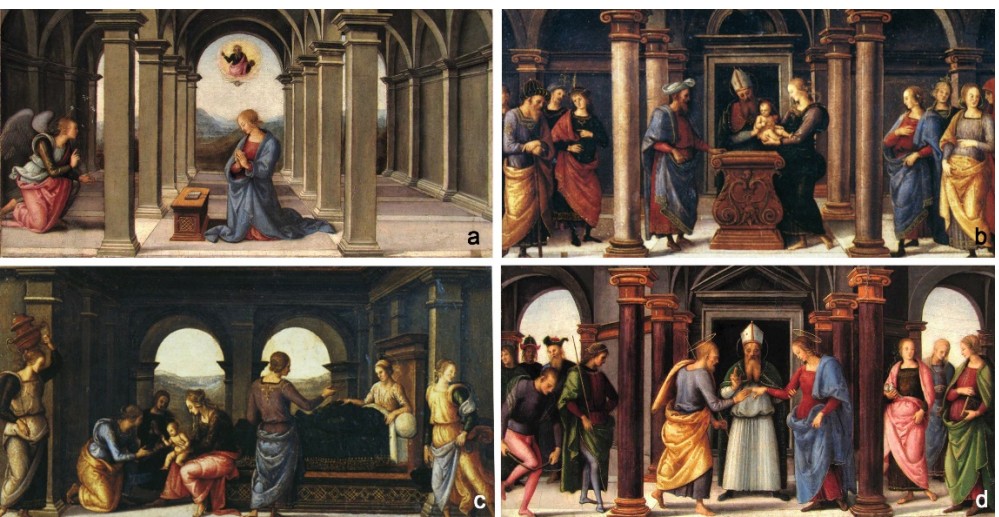

**Figure 3.** Pietro Vannucci known as Perugino (and Raffaello?), The Fano Altarpiece, 1497, church of S. Maria Nuova, Fano: (**a**) The Annunciation; (**b**) The Presentation; (**c**) The Stories of the Virgin; (**d**) The Marriage.

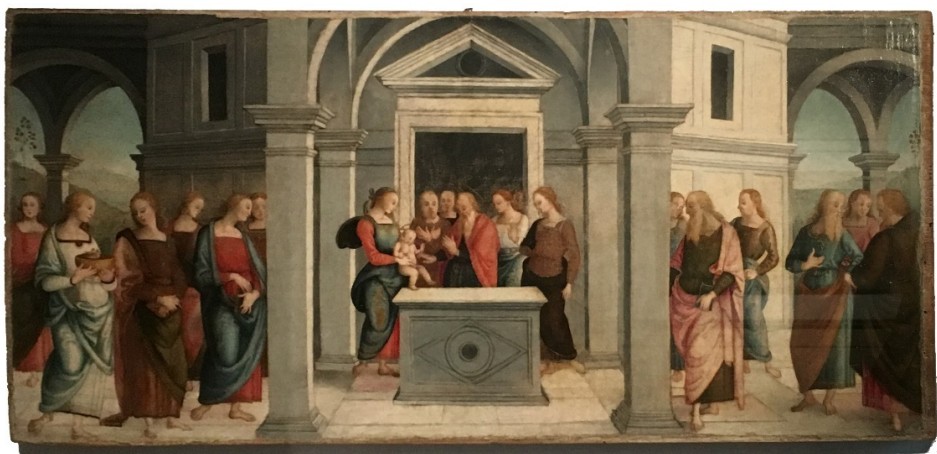

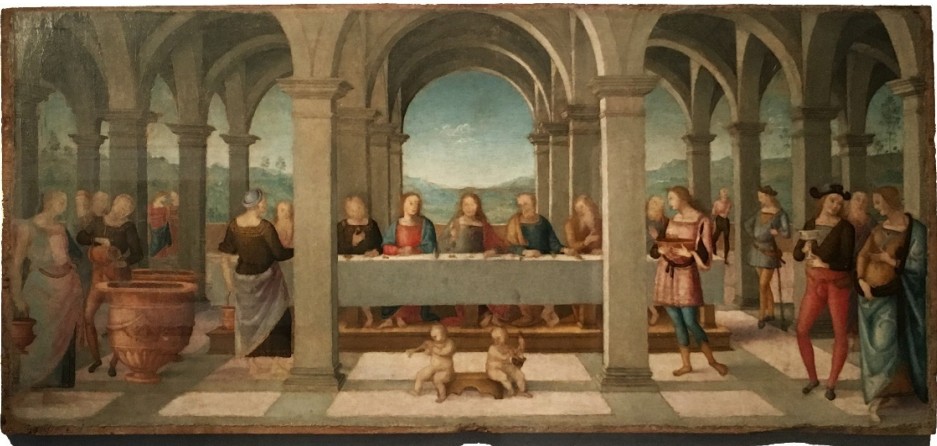

**Figure 4.** Pietro Vannucci known as Perugino (and Raffaello?), Polyptyc of St. Augustin, 1502–1523, Galleria Nazionale, Perugia: The Presentation to the Temple (**above**); The Marriage at Cana (**below**).

Raffaello also had the opportunity to develop his architectural models on behalf of the Sienese Bernardino di Betto Betti, known as Pinturicchio (1452–1513), another of Perugino's pupils. The fresco cycle of the Piccolomini Library, Siena, was designed in a rectangular hall as a sequence of 12 huge 'windows' (4 on the longer sides and 2 on the shorter ones). Except two real windows, the others are virtual openings on painted scenes of remote places. While the architectural pillars and arches that separate the 'windows' are depicted in perspective with the horizon line at the eye level in order to suggest an illusory expansion of the hall, the painted scenes inside them present a higher horizon line, accentuating the effect of detachment and movement from different realities.

The pseudo-basilica building on a seven-step plinth depicted in *The Coronation of Enea Silvio Piccolomini*, which Frommel attributes to Raffaello, represents the Imperial Palace in Aachen, Germany, where the coronation took place. At the same time, it recalls one of the hypostyle halls painted by Perugino seen from the outside (Figure 5). The ground floor presents five naves with three deep bays. The cross vaults set on round arches rest on pillars, while bands of the white stone border the floor squares. Despite being conceived for a painting, the painter designed it as an architect would do: he reduced all the elements to simple proportions, used cross-shape pillars to support the vaults better; added volutes to bear the lateral thrusts of the upper barrel vault; showed the first steps of a staircase.

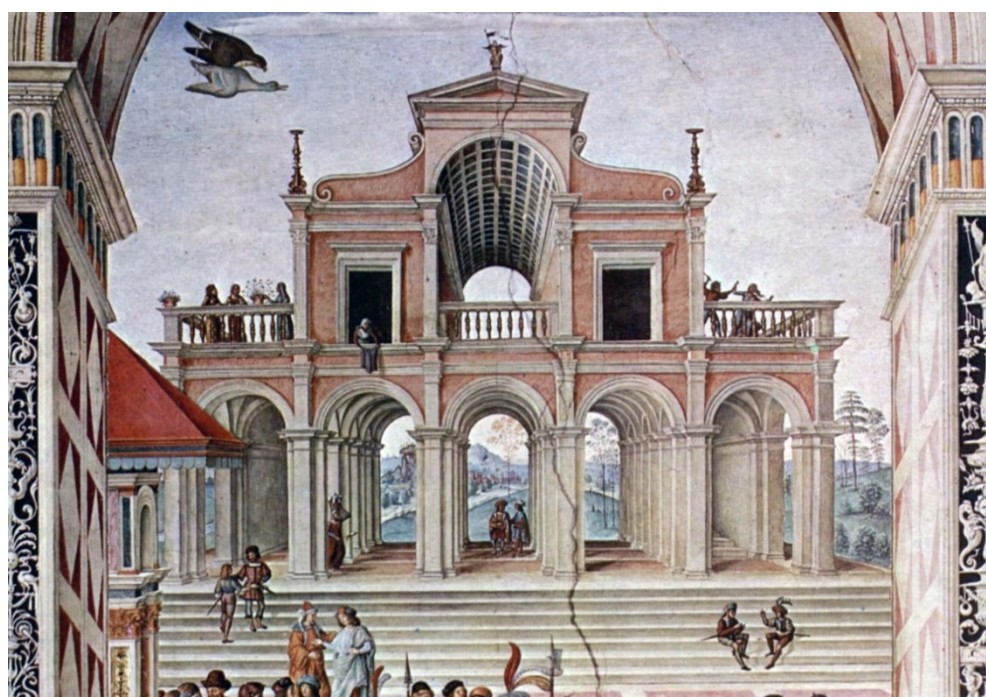

**Figure 5.** Pinturicchio (and Raffaello?), Detail from The Coronation of Enea Silvio Piccolomini, 1502–1507, Piccolomini Library, Siena.

In *The Annunciation* of the Pala degli Oddi (1503), Raffaello extended the gallery of Perugino's Fano Altarpiece to four naves, replaced the square pillars with Corinthian columns (Figure 6a), and used the central pillar to divide the Archangel's 'celestial' space from Madonna's 'earthly' space (Flint 2014). The octagonal temple with canopies in *The Presentation* of the Pala degli Oddi presents Ionic columns and pillars (Figure 6b), while the 16-side temple of *The Marriage* (1504) at the Brera Pinacoteca is ringed by a peristyle on free columns connected by volutes to the upper drum which supports a hemispherical dome with a lantern.

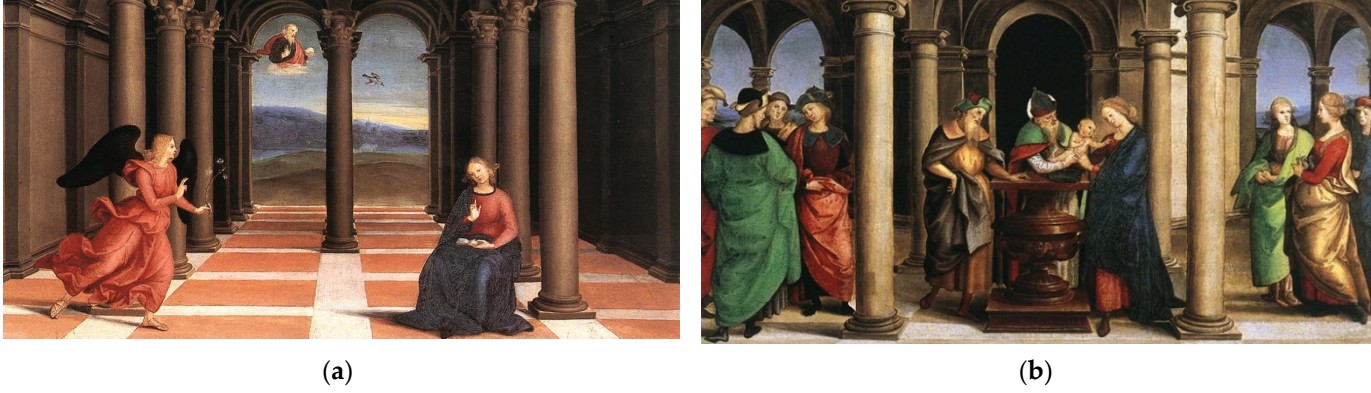

**Figure 6.** Raffaello, Pala degli Oddi, 1503, Vatican Art Gallery, Rome: (**a**) The Annunciation; (**b**) The Presentation.

While growing from a simple assistant into a celebrated master, Raffaello experimented with many different architectural typologies, gradually learning how to quote current buildings and antiquities, such as the Pantheon-shape half-dome on Corinthian columns in *The Madonna of the Baldacchino* (1506–1508; Figure 7a). Perugino's hypostyle hall is radically reconfigured in *The Healing of the Lame* (1515; Figure 7b) that Raffaello painted when he had already taken the place of Bramante as the main architect of the *Fabbrica* of St. Peter. Visible both in the tapestry and in the painted preparatory cartoon, *The Healing* shows a portion of two pronaoses composed of twisted columns and crowded with figures that seem almost imprisoned.

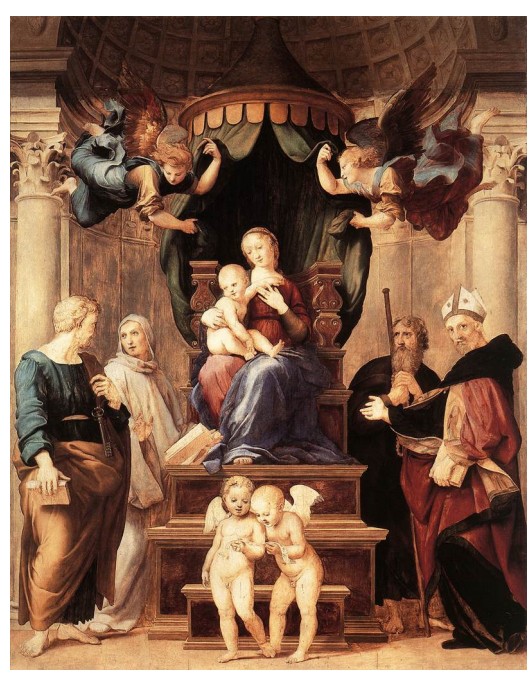

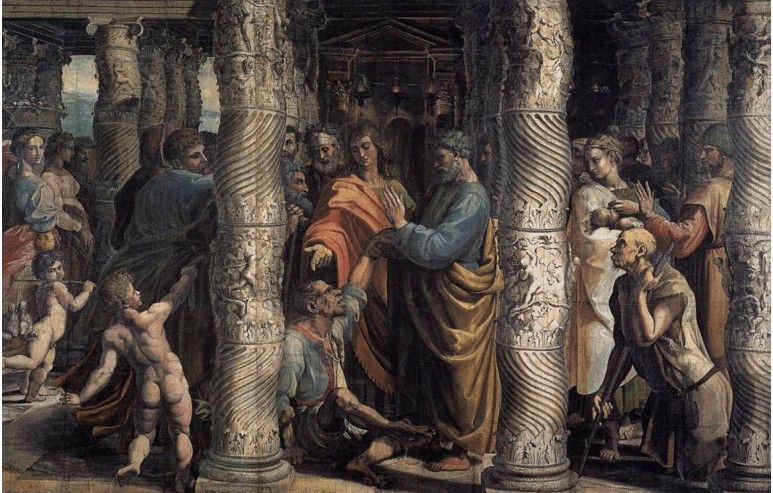

(**a**)  (**b**)

**Figure 7.** Raffaello, (**a**) The Madonna of the Baldacchino, 1506–1508, Galleria Palatina, Florence; (**b**) The Healing of the Lame, 1515, Victoria and Albert Museum, London.

On the contrary, the architecture painted years before in *The Disputa* (1509; Figure 8), a famous fresco in the Stanza della Segnatura, Vatican Palaces is evoked mainly by the figures' disposition. The figures are arranged on three levels and combine with the altar and the step below to suggest the interior of a church. The lower level is characterised by the pyramidal configuration of the altar above four steps and a chequered floor, and a

white stone cube raising beyond the heads of the standing figures. A semicircular choir of clouds occupies the intermediate level with seated figures and the enthroned Christ in the middle. The upper level, as Marcello Fagiolo (2020) noted, is topped by a sort of half-dome set on the ring of clouds around the Creator and outlined by ribs of angels that seem to allude to the Pantheon's dome.

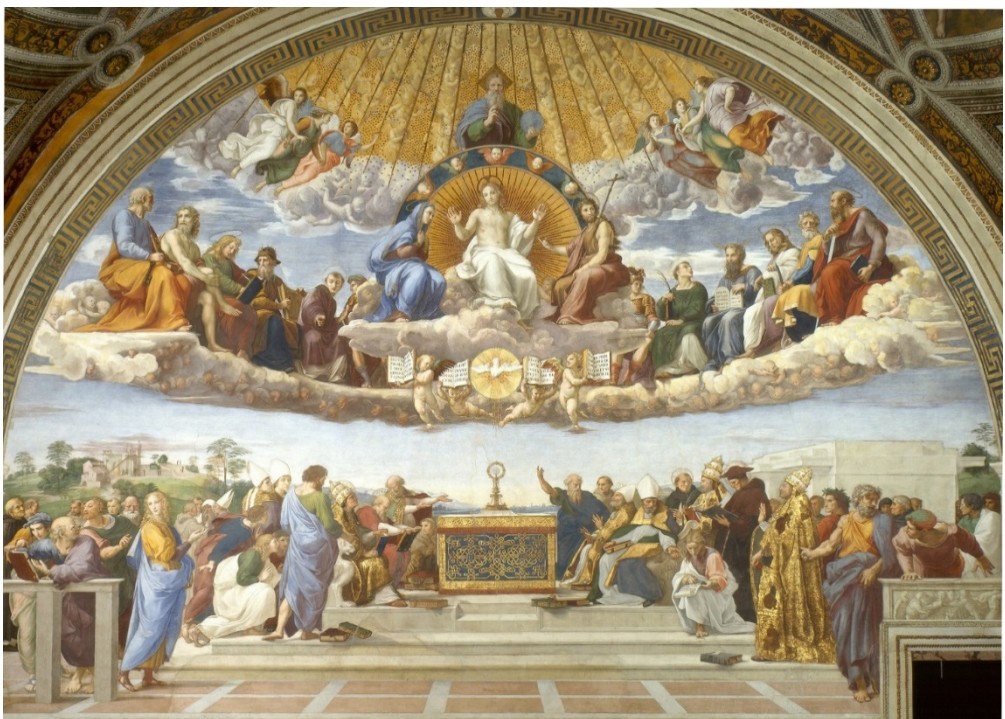

**Figure 8.** Raffaello, The Disputa, 1509, Vatican Palaces, Rome.

## 3. Perspective Analysis and Restitution

The perspective analysis was initially applied to Perugino's works that share a structure on pillars arranged in a regular grid, and then it was extended to others by Raffaello. The perspective restitution process, carried out on high-resolution digital reproduction of a frontal photoshoot of the works, was limited to a selection of Perugino's and Raffaello's works chosen to show the quantitative and qualitative range of the architectural solutions. The scale reported in the drawings results in an approximate measurement based on the size of the painted figures. In some restitutions, the directly visible part of the architecture was integrated by elements logically deduced by conjecturing a symmetric configuration.

The perspective restitution is based on the hypothesis that the floor is designed as a grid of squares. It was thus found that the perspective structure of Perugino's works is always rigorous; moreover, the three-dimensional modules of the structures refer to a cube, while the height of the pillars varies from about two meters to four and over, such as in *The Last Supper of Fuligno* (Figure 9). Other results concern the pillars. The capitals and bases present only two formal variations, and the height of the shaft is from 6.5 to 8 times the base side.

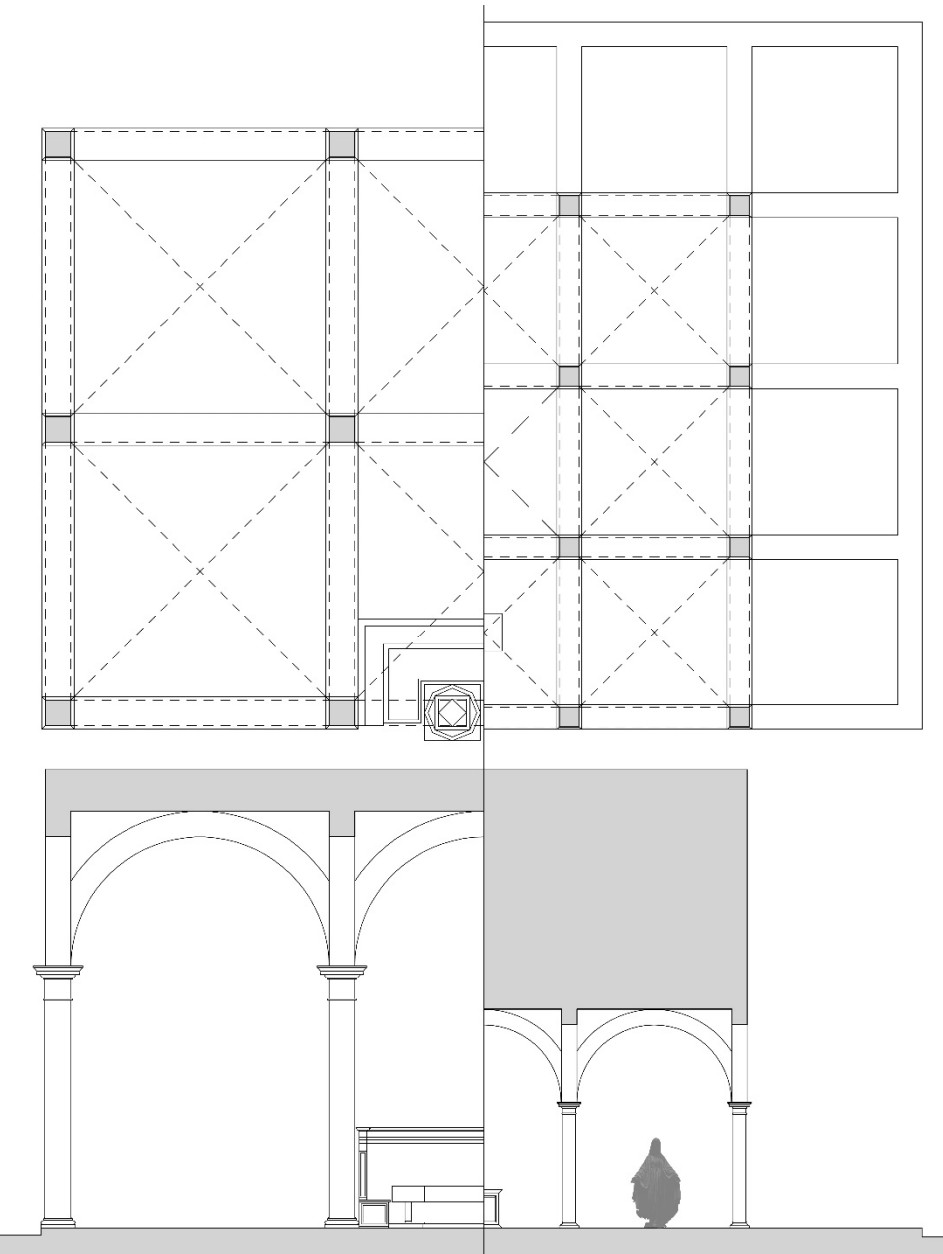

**Figure 9.** Dimensional comparison between half-plan and half-section from the restitution of Perugino's Last Supper of St. Onofrio, 1482, and The Annunciation of Fano, 1489.

After the perspective restitution, the ideal point of view and the visual cone were added to plans and elevations to highlight the strictly visible portion (in light grey) and some geometric-perspective parameters (Figure 10). In the case of Perugino's works, the visual field ranges from 23° (*The Annunciation* in Fano) to almost 50° (*The Wedding at Cana*). The lower frame of the scene generally coincides with the band that binds the pillars or is marked by unique decorations (from *The Last Supper of Fuligno* to *The Wedding at Cana* of the Polyptych of St. Augustine) to underline the fundamental line of the perspective construction and, at the same time, the edge of the virtual stage. The point of view can also be very low, such as the knee in *St. Sebastian*, to amplify the majesty of the figures and the visibility of the vaulted surfaces. At the same time, the point of view is generally distant; this favours a correct and proportionate representation of the human figures (almost in orthogonal projection) but prevents a bodily engagement in the pictorial space. The field

of view is generally narrow to avoid the peripheral aberration of the central perspective ([Trevisan 1999]).

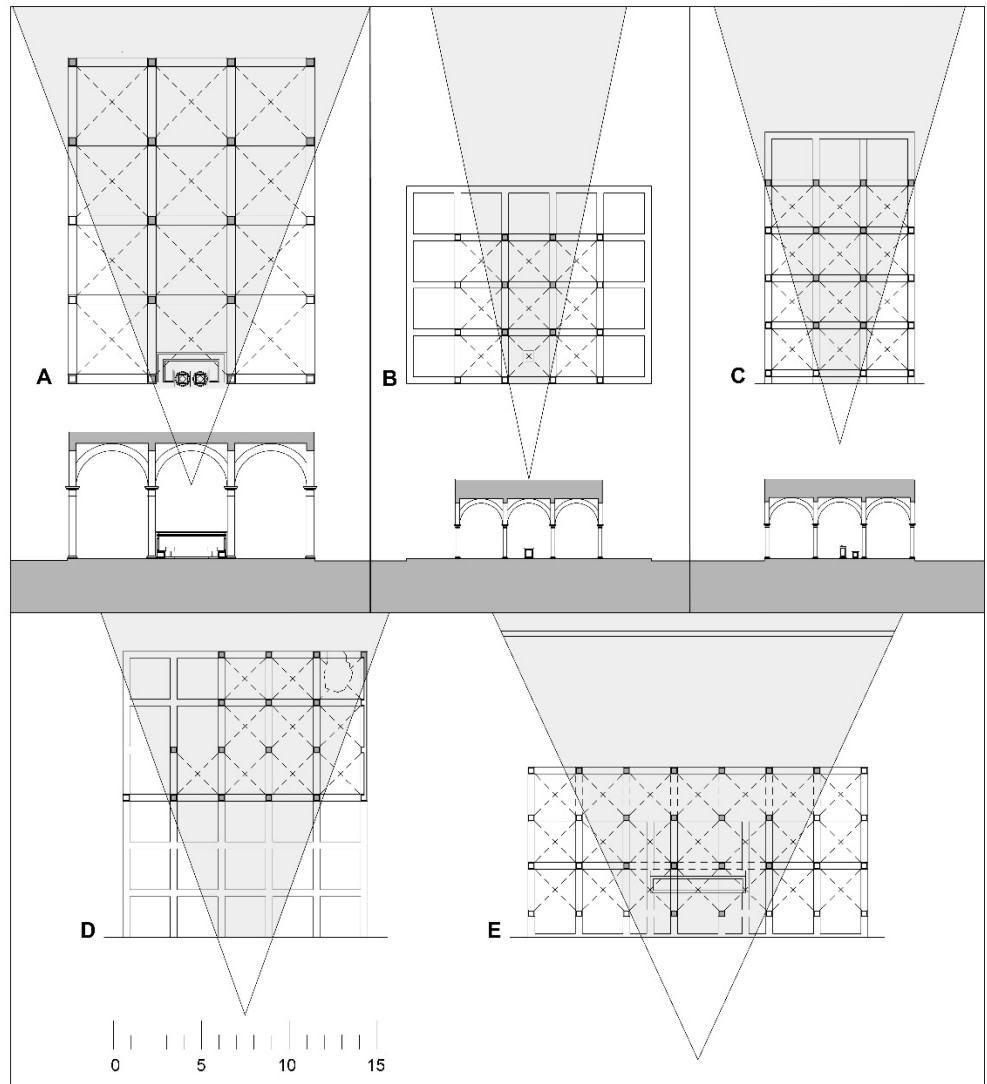

**Figure 10.** Perspective restitution in plan and section with the visual cones from Perugino's works: (**A**) The Last Supper of St. Onofrio, Florence; (**B**) The Annunciation of Fano; (**C**) The Vision of St. Bernard; (**D**) The Nativity of Albani-Torlonia Polyptic; (**E**) The Wedding at Cana in St. Augustin.

The fictive quality of Perugino's hypostyle halls is enhanced by other elements; they look like ephemeral wooden structures painted with homogeneous colours, occasionally placed directly on the grass. Unlike the figures, the pillars are often devoid of casting shadows, and some of them are pragmatically removed to show the figures behind them; moreover, despite being correctly painted and foreshortened in the pictorial space, Perugino's figures seem almost indifferent to the painted architecture. On the contrary, a stronger interaction between figures and architecture is evident in Raffaello's paintings since *The Annunciation* of the Pala degli Oddi. Although the mathematical plan still resembles Perugino's architectural models (Figure 11), the light and colour of the figures seem to echo those of the columns.

The perspective restitution on *The Madonna of the Baldacchino* involved both the architectural scenario and the piece of furniture, the tall wooden throne with the hung baldachin. While the upper part of the elevation was completed by assuming that the half-dome mimics the Pantheon's dome, the restitution raised questions about the actual form of the

niche and the half-dome, which seems to be partly impressed into the wall and partially supported by a pair of Corinthian columns (Figure 12a).

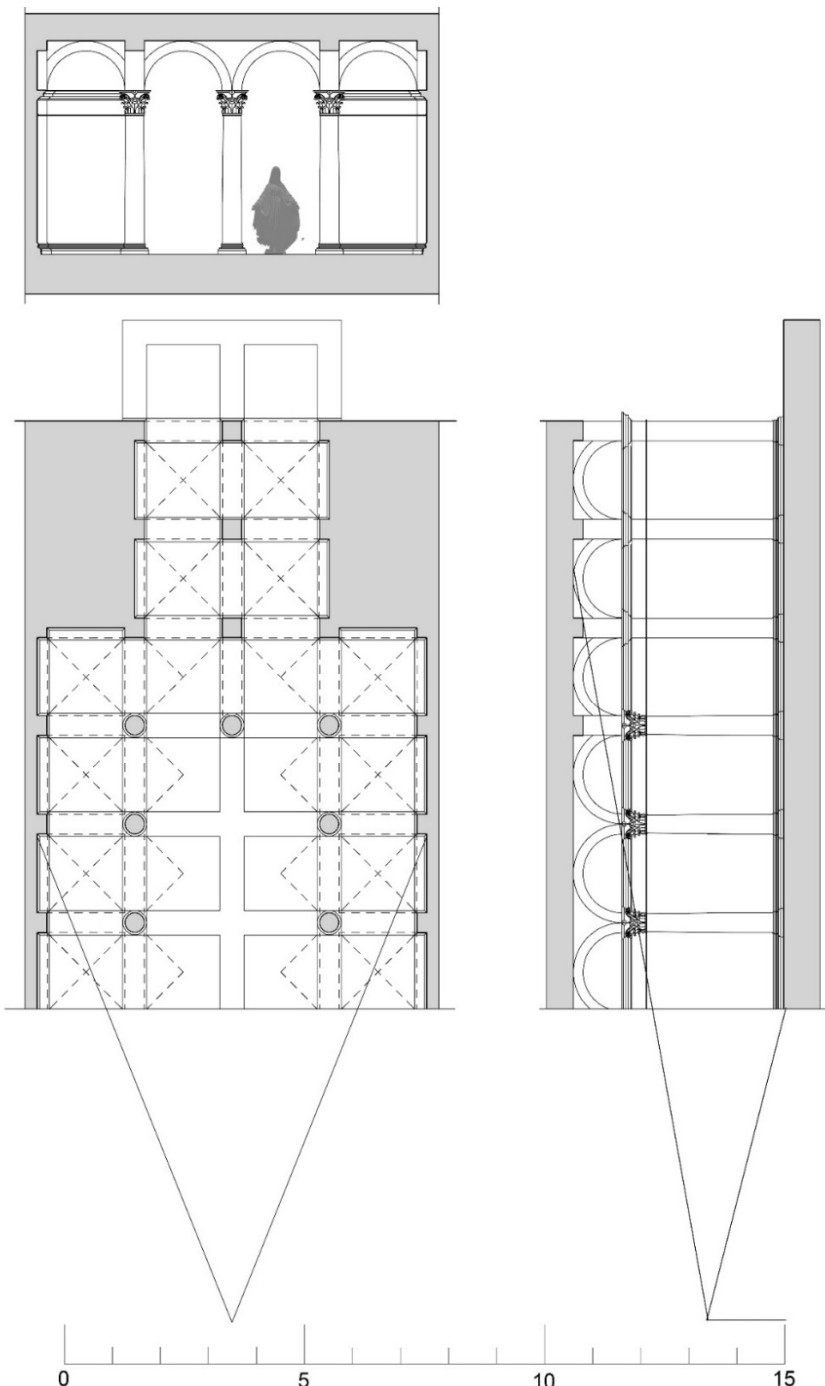

**Figure 11.** Perspective restitution in the plan, transversal section (**above**) and longitudinal section (**right**) with visual cones from Raffaello's *The Annunciation from the Pala degli Oddi*, 1503.

In *The Healing of the Lame*, the pillars look as important and corporeal as the figures do. The perspective restitution provided a disproportionate architecture, conceived as a pure scenography but composed with carefully designed elements, such as the decorative twisted columns, whose grooves 'compete' with the drapery of the garments. Raffaello chose a viewpoint at the figures' eye level; however, it is not central but eccentric in order to show a part of the side pronaos, as well. Raffaello chose a distant point of

view that produces a telephoto effect, optically 'compressing' both the columns and the figures between them, while only a small portion of the horizon is still visible on the left (Figure 12b).

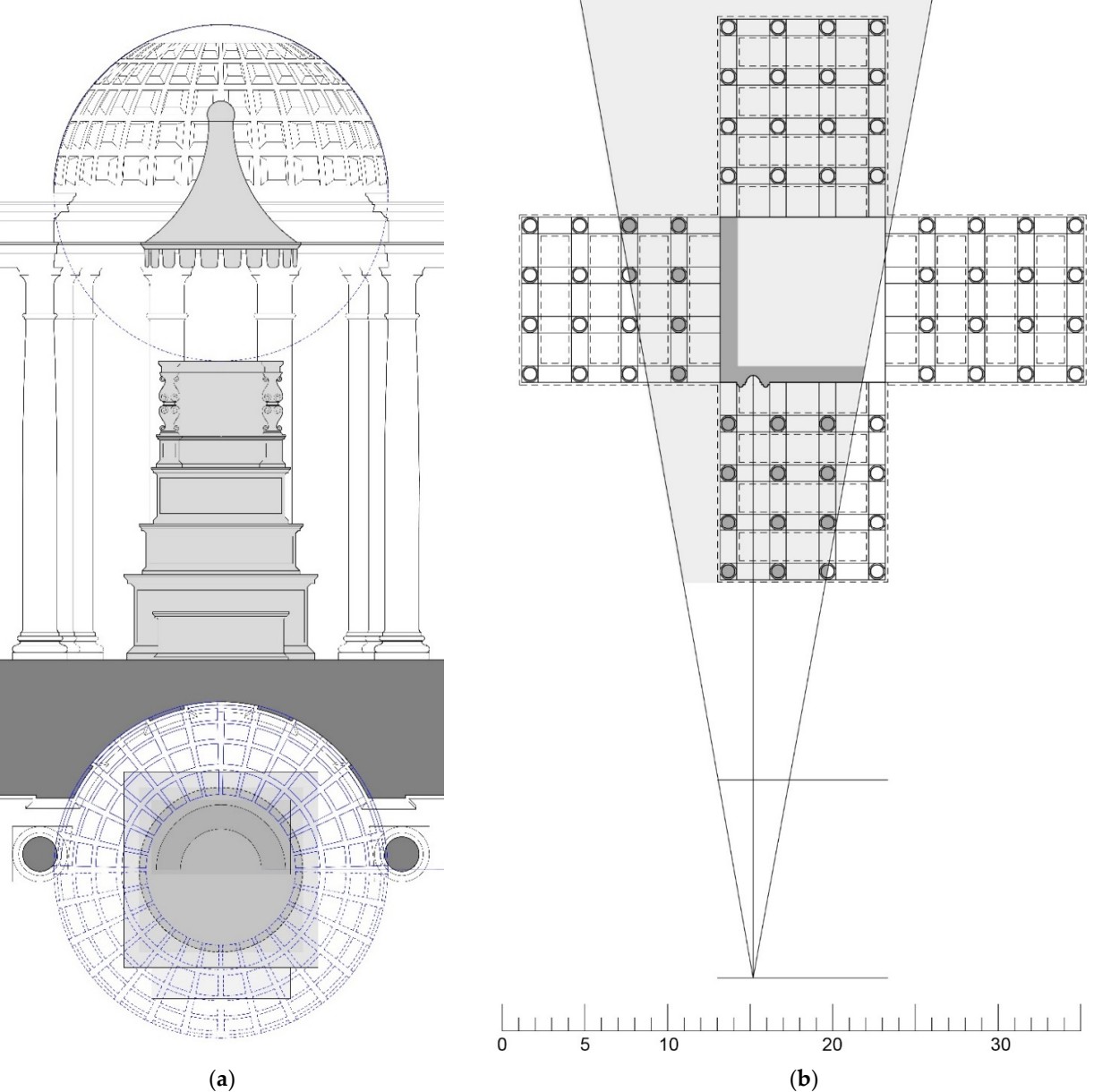

(**a**)                                                                           (**b**)

**Figure 12.** (**a**) Perspective restitution in plan and elevation from Raffaello's *The Madonna of the Baldacchino* with the throne in light gray; (**b**) Perspective restitution in plan with the visual cone from Perugino's *The Healing of the Lame* with the effective visible part in light gray.

Raffaello's *The Dispute*, which has been the subject of several studies (Shearman 1972; Kuhn 1990; Winner 1993) concerning its symbolisms and relationships with the other frescoes in the room, requires some preliminary considerations, which can also be extended to *The Coronation of Enea Silvio Piccolomini* and its mathematical design (Figure 13). Both the scenes, *Dispute*'s and *Coronation*'s, are placed above a decorated base, which was presumed to be covered by the library shelves. The base line of the perspective scene is placed at the eye level of a standing beholder, simulating the height of the floor of a theatrical stage, while the horizon line is much higher. In this way, the scene cannot produce an illusionistic effect, which occurs when the eye level and perspective horizon line coincide (Figure 14).

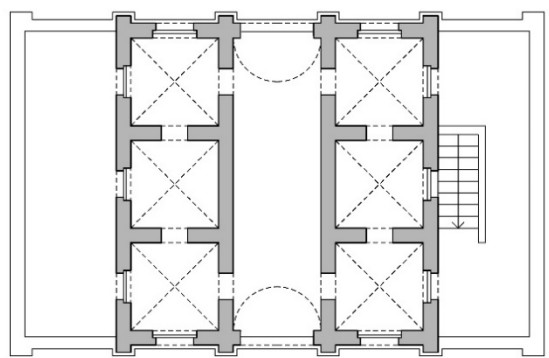

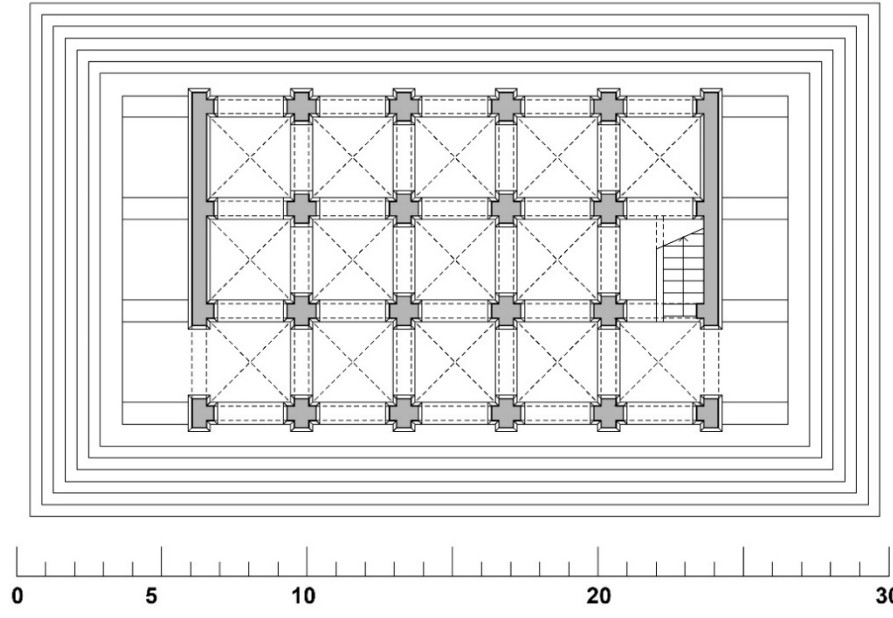

**Figure 13.** Perspective restitution in the plan of the ground floor and first floor from Pinturicchio (and Raffaello?)'s *The Coronation of Enea Silvio Piccolomini*, Piccolomini Library, Siena.

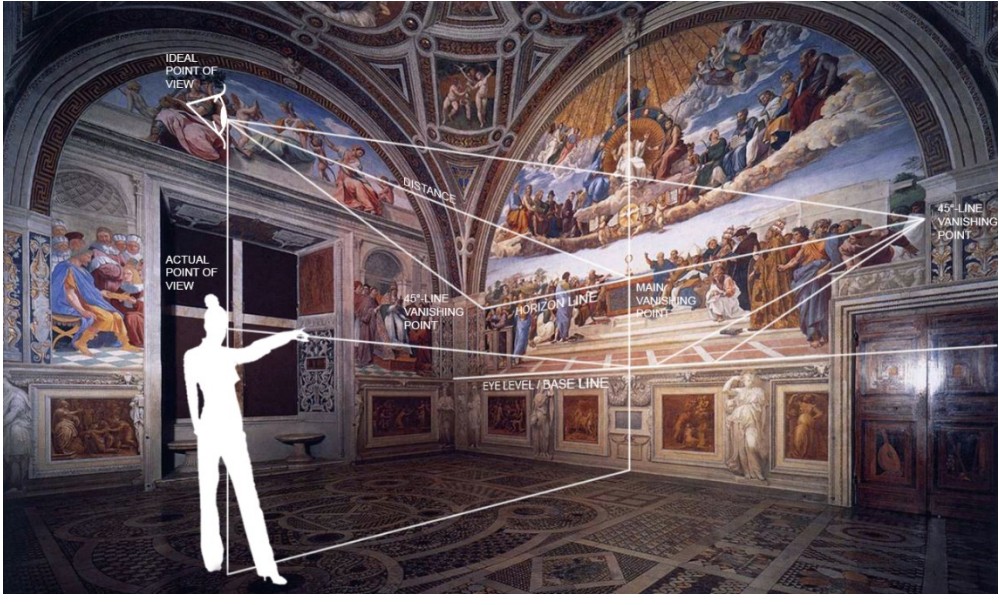

**Figure 14.** General perspective structure of Raffaello's *The Disputa* and the beholder in front of it.

The three levels of figures in *The Dispute* appear to be hinged on the vertical axis of the monstrance upon the altar. The perspective analysis individuated the central vanishing point at the base of the monstrance itself, where the horizon line passes. Both "earthly" and "celestial" painted elements are ruled by this vanishing point.

To establish the shape and position of the altar and the steps below, the floor partitions were initially assumed as square. When extended, the diagonals of these partitions are expected to individuate two secondary vanishing points on the horizon line. While in *The Coronation of Enea Silvio Piccolomini*, this procedure individuates the secondary vanishing points in actual points on the wall at the left and right of the fresco, in *The Disputa*, the result is unsatisfactory as both lateral vanishing points are beyond the side walls and unusable; moreover, the ideal point of view refers to an impossible position beyond the back wall. Although the role of the point of view will be later considered in a more dynamic way (Gay 2014), at the beginning of the 16th-century placing it consistently with the beholder's position is one of the main goals of the artists even when they have no illusionistic intent.

Conversely, by assuming two contiguous floor partitions as a square and extending the diagonals, two secondary vanishing points of *The Disputa* were individuated in actual points of the horizon line before the corners of the room. According to this schema, the ideal point of view is about 4.50 m from the fresco, where an observer can stand and look at the scene. At the same time, the observer's eye is not as high as the ideal point of view and indeed, no illusory effect can be activated.

The conjectural perspective structure is confirmed by the altar, which is revealed to be modulated based on a square grid. Five square modules in depth allow identifying, in sequence: the strip of flooring; the first two steps; the edge of the upper step in front of the altar; the size of the altar itself; the edge of the upper step behind the altar (Figure 15). When imagining a similar articulation for the back of the altar, a symmetrical composition of 7 modules results. As the first step is 9 modules wide, a 7:9 rectangular altar structure results, while the lack of direct connections between it and the higher levels prevents the geometric restitution of the whole pictorial space.

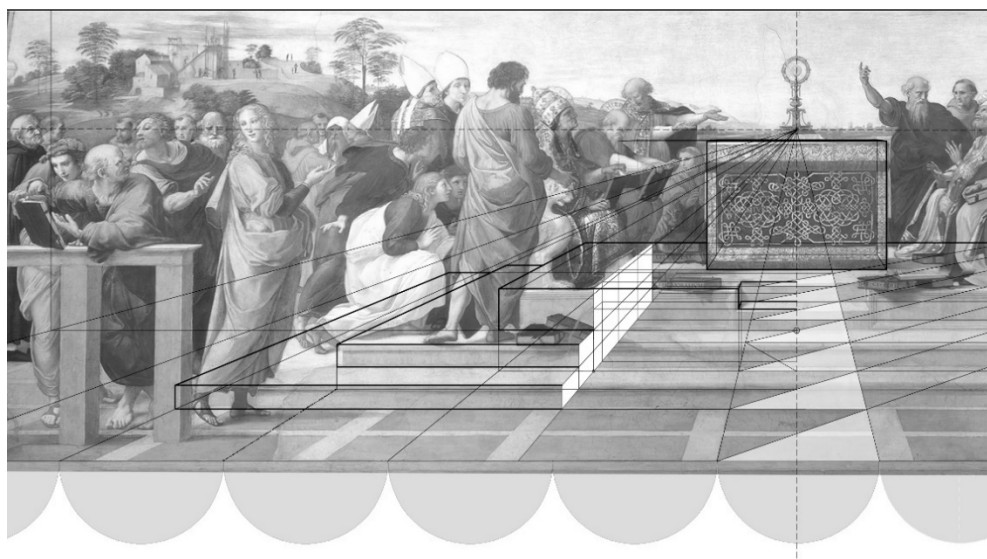

**Figure 15.** Reconstruction of the perspective structure of the steps under the altar of Raffaello's *The Disputa*.

In a second stage, the restituted plan, was used to insert a triumphal arch into the pictorial space of the fresco. According to Fagiolo's interpretative hypothesis, the white stone cube on the right, which is missing a small block in the corner, constitutes the lower part of the pylon of a triumphal arch. Unfortunately, the figures hide the lower part of the block, and its position cannot be restituted with certainty. At the same time, it is as wide as

the altar, as if the artist intended to suggest that the two elements are aligned. Starting from this and other hypotheses, a perspective restitution was attempted. The pylon resulted in an asymmetrical plan, which can be interpreted as both an indication of the position of the pilasters and a slight imprecision by the painter. In this sense, the block was interpreted once as the left pylon and once as the right pylon of the hypothetical Arch.

Starting from the restituted pylon, a Triumphal Arch was designed in plan and elevation (Figure 16); it was inspired by ancient models, such as the Arch of Titus in Rome and the Arch of Trajan in Ancona, and the arch painted by Raffaello himself in the coeval *The School of Athens*, behind the figures of Plato and Aristotle. The Arch was then translated into a three-dimensional digital model. Two different renderings were produced according to the point of view and perspective structure of the fresco and to the two hypotheses on its position. Finally, the perspective views of the Arch were inserted into the digital reproduction of *The Disputa* over the block.

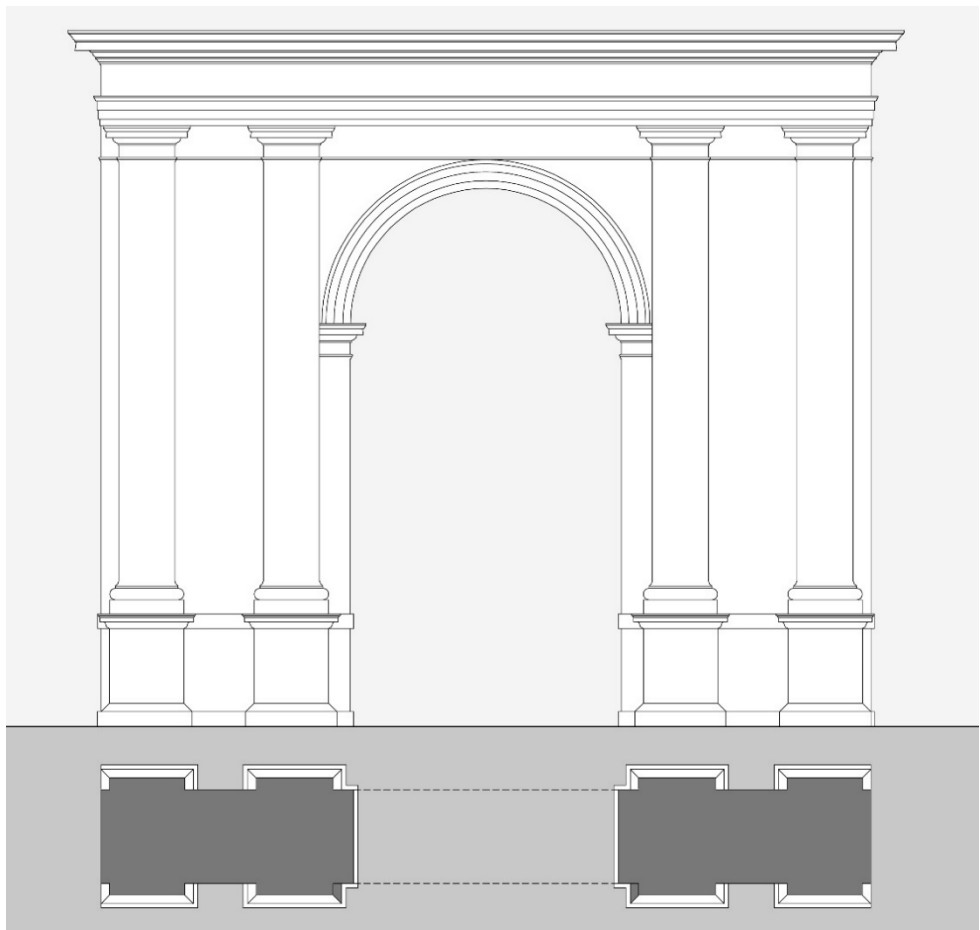

**Figure 16.** Plan and elevation of the Triumphal Arch to be inserted in Raffaello's *The Disputa* after the restitution plan of the pylon.

The resulting digital photomontages were used to assess the most plausible position of the conjectural Arch, using the model to offer a visual development of what the pylon merely suggests and to evaluate the consistency of pictorial space and the consequences on the whole work (Figure 17a,b).

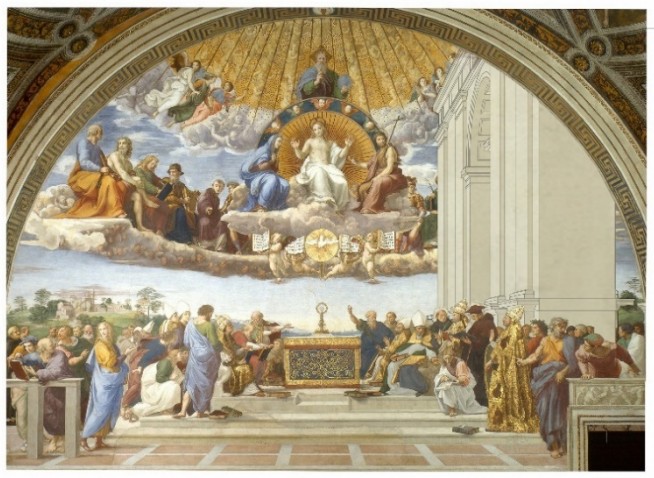 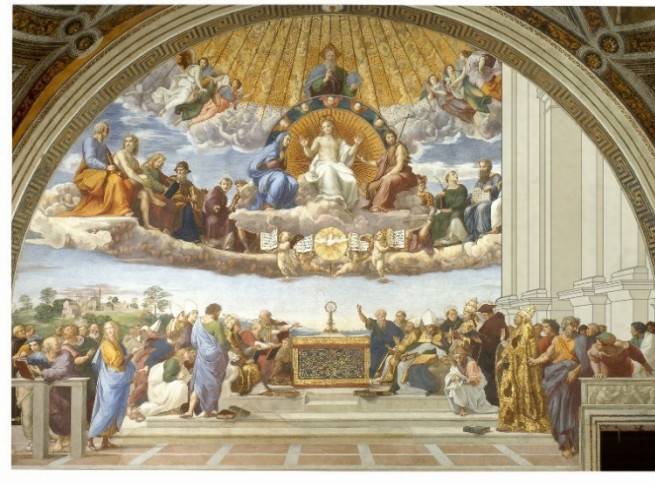

(**a**)     (**b**)

**Figure 17.** Digital Photomontage of the Triumphal Arch in Raffaello's *The Disputa*, considering the stone block as the right pillar (**a**) or the left pillar (**b**).

## 4. Considerations

As written in the *Introduction*, the perspective analysis and restitution can provide data that foster considerations on cultural aspects of the work and may concern both the medium, the perspective painting, and the object, the architecture portrayed. Perugino's hypostyle architecture is inspired by what was being built in Florence, a city that Perugino had been frequenting increasingly intensely (in 1493, he married the daughter of Luca Fancelli, a former collaborator of Leon Battista Alberti and first architect of the Duomo) and in which he opened a shop in 1504.

In figurative terms, his painted architecture recalls collective urban structures that can generically represent a public palace and provides a visual reference to orient the beholder in the pictorial space. A permeable and transparent development of the opaque perspective box adopted by the previous generation of painters, it proves an abstract and flexible scenography that can simultaneously represent the interior and exterior, a room and a square; in a word, it reveals an 'adaptive' device, such as certain software, to the specific size of the frame, the narrative program, and the figures it is designed to host.

In perspective terms, Perugino's architecture constitutes the three-dimensional reflection or transcription of the square grid that was preliminarily traced and foreshortened as prescribed by Leon Battista Alberti and Piero della Francesca. In this sense, the metal bars chaining the vaults in the *Madonna with Child and Saints* in the church of S. Maria delle Grazie, Senigallia, manifest the three-dimensional cubic matrix. Following the suggestion of the Adam hypostyle evoked by George Hersey (1976) about the *Pythagorean Palaces*, Perugino's hypostyle structure could be interpreted as a sort of invisible, mathematical palace evoked by the architectural orders.

Retrospectively, a compositional methodology seems to emerge when confronting the results of this analysis with the organization of Perugino's shop. For many years, Perugino was one of the most relevant artists in Italy, with a vast number of commissions to attend at the same time in different places; he needed to organise his modus operandi to let his talented collaborators help him efficiently but, at the same time, to control them and keep the quality high, as well. For example, his works reveal frequent reuse of models for the figures, which could reduce the composition to a sort of montage. In this sense, the organisation of the painting as a mechanical superimposition of three layers—the landscape in the background, the architecture, and the figures—can be interpreted as an important working methodology, a justification for the infrequent interaction between the figures and architecture, and also an unpredictable link with the current software for digital painting and animation.

Raffaello's *architectura picta* initially evolved in the wake marked by Perugino, both in typological and perspective terms. Frommel (1984, p. 16) stated that, as late as 1505, the artist did not appear interested in revolutionizing or innovating architecture but rather, he "seeks what completes and perfects the world of Perugino". From the examples considered here, it is possible to conjecture that he contributed to the consistency of the *architectura picta*, as perhaps identified in Siena and more certainly in the colonnade of *The Annunciation* of 1503. Both at the scale of the single element and the whole architecture, Raffaello translated Perugino's angular surfaces into continuous surfaces capable of fading light and colour as much as the robes of the figures; he gradually abandoned the master's set of cubic frames as a response to his early studies on antiquities, the architectural orders, and their proportions, and this pushed his painted architecture closer to the architecture discussed, surveyed, and designed in Rome, especially in the context of the new Basilica of St. Peter and the circle of Donato Bramante, a master of perspective too (Camerota 2001).

At the same time, a stronger relationship between human bodies and architectural bodies, in terms of physical interaction and chiaroscuro and chromatic effects, is largely appreciable. Bussagli (2020, p. 107) suggests that Raffaello was influenced by the diffusion of the Vitruvian ideas and his analogy between a building and the human body, which was indicated as an *exemplum* to be followed by architects. Surely, while Perugino used the three-dimensional frame to separate and organise the figures into groups, in continuity with the figurative model of the medieval polyptych, Raffaello used the columns of *The Healing of the Lame* to stage an anti-theatrical space designed to hide more than to reveal or order. The figures interact closely with the structure, its interstices, and its cast shadows to produce a claustrophobic density of bodies; this criterion of "occupation" and "manifestation" of space, which breaks with Perugino's compositional canon, is explicitly appreciated in *The Madonna of the Impintage* (1514) where Robin Evans (1997, p. 59) noted that "the figures are not so much composed in space but gathered together despite it."

The figures were a fundamental factor in involving the viewer in the pictorial space. As explained by Kornelia Imesch-Oehry (1998), Alberti's "open window" visual model did not envisage an illusory connection between the pictorial space and the beholder's space, quite the contrary. Alberti's architectural conception is fundamentally anti-illusionistic, as the censorship he imposed on architects against decorated models, and perspective drawing testifies; he considered perspective as a structure that orders the figuration of space to serve the *historia* and provide an internal spatial coherence to the image. In this sense, the Albertian window was designed to show a world in perspective, as seen by the main figures in the pictorial space. Unlike the illusory Masaccio's *Trinity* in the church of S. Maria Novella, Florence, in Alberti's opinion, the height of the perspective horizon line was not determined according to the position of the beholder. Instead, painted figures and contemporary characters were called to act as mediators that invite the beholder to enter the pictorial space, such as in a movie.

In the years of Perugino and Raffaello, Alberti's approach was occasionally combined with the illusionistic one. For example, in the case of the basement and the frames of the Piccolomini Library, Siena, and the Stanza della Segnatura in the Vatican Palaces are painted in perspective according to the eye level, as a virtual extension of the actual room, while the scenes present a higher horizon line and require a mediated involvement.

The restitution of *The Dispute* also constitutes an opportunity to reflect on the relationship between the geometric space resulting from the perspective restitution and the mental space suggested by the painting, indirectly referring to the artist's intentions (Figure 18). Like most artists, the main goal of Raffaello was to produce an image consistent with the figurative and symbolic program; therefore, it seems plausible that he sought and found a compromise between the design of the pictorial space, the perspective medium, and the geometric conditions of the room. In particular, the artist used the perspective construction to suggest a greater virtual depth of the pictorial space and a greater distance.

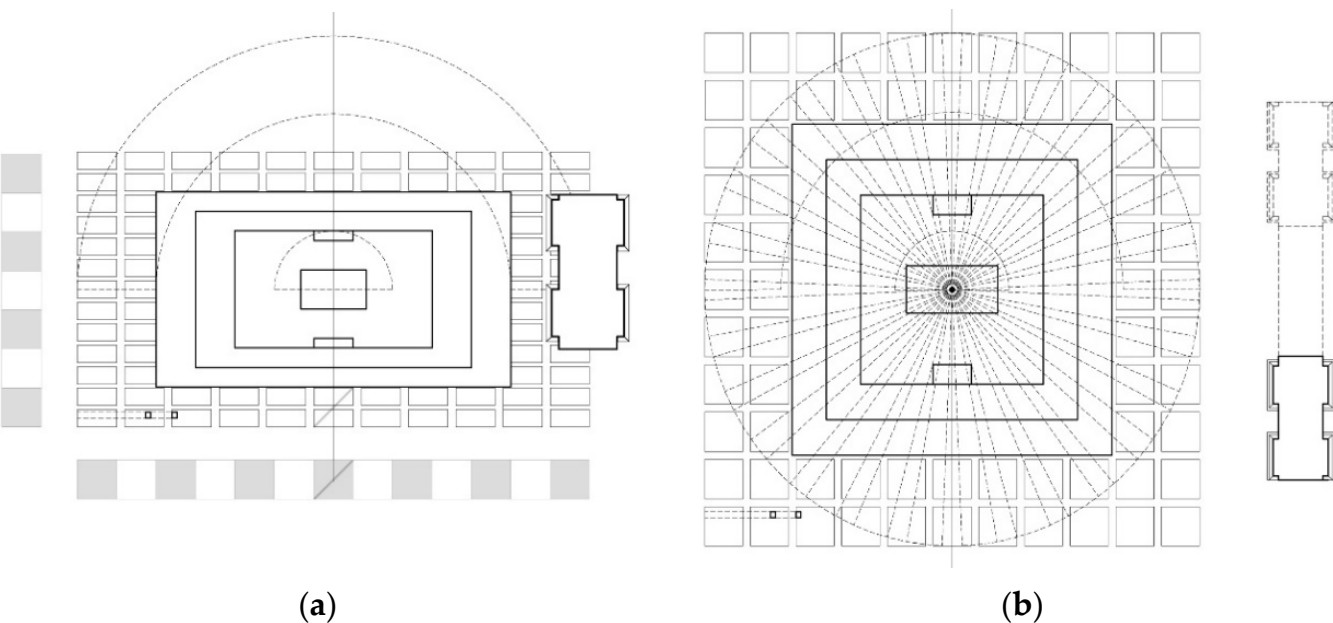

**(a)** **(b)**

**Figure 18.** Comparison between the actual plan after the perspective restitution (**a**) and the plan suggested by the vision (**b**) of Raffaello's *The Disputa*.

Moreover, the organisation of the ground floor infers the interpretation of the upper levels, which are virtually connected to the altar configuration but actually detached and can be visually investigated only through the size and position of the figures. In this sense, the elementary and measurable geometry of the lower level is transfigured into the "atmospheric" divine materials of the upper levels. The semi-circular choir seems to be virtually sectioned by the vertical plane passing through the monstrance itself, but the size of the figures suggests that it is closer to the beholder; this aspect was further tested through the insertion of the triumphal arch, which, being an extraneous element, highlights the spatial inconsistencies, especially in the relationship between the altar and the figures of the second level. If the Arch had been entirely painted, it would have compromised the visibility of the right section of the upper levels, covering many of the seated figures and certainly stealing the scene from the "invisible" cathedral set up in the natural landscape; this could be prevented by moving the figures before the Arch, eventually breaking the geometric coherence of the whole system. Conversely, when considering the painted pylon as the left one, most of the Arch would be closer to the observer and would be cut by the decorated arch that frames the fresco, eventually suggesting that the Triumphal Arch is placed on the axis of the altar.

## 5. Conclusions

The process of analysis and perspective restitution, occasionally complemented by the three-dimensional modelling of the conjectural Triumphal Arch of *The Disputa* (Figure 19), was applied to study the architecture painted by Pietro Perugino and Raffaello Sanzio, his pupil, between the 15th and 16th centuries. The drawings in plan, elevation, and section after the restitution of the painted buildings are the primary results; they allow us to measure the pictorial space and architecture, compare them with actual projects and buildings, and testify to their perspective coherence.

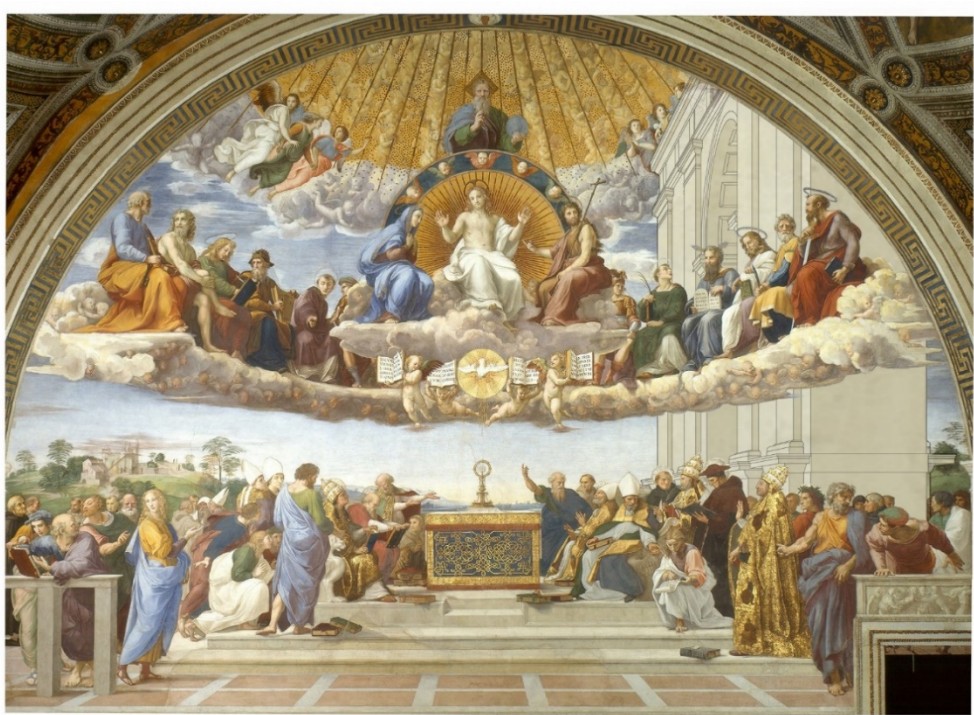

**Figure 19.** Final version of the digital Photomontage of the Triumphal Arch in Raffaello's *The Disputa* considering the stone block as the right pillar.

On the other hand, the secondary outcomes involve a multiplicity of aspects. Concerning the general composition, a painting procedure resulting from the overlapping of three autonomous layers has emerged as a response to the organisation of Perugino's workshop, while the hypostyle hall provided an adaptive device flexible to narrative and figurative requirements. Initially, Raffaello aligned himself with the master's iconographic model, enhancing the constructive aspects of the architecture. Gradually, he was increasingly involved in antiquarian studies and the theatrical, fictive features of perspective paintings. In this sense, he introduced columns instead of pillars to accentuate the sensuality of the surfaces and increased their interaction with human figures, presumably also to amplify the beholder's involvement in the pictorial space, but this also had the consequence of making the invisible plan grid less evident and of giving precedence to the visual effect over the architecture geometric and constructive consistence. The analysis of *The Dispute* also demonstrates Raffaello's technical ability to circumvent the operational constraints of perspective and to mediate between what is suggested by the image and what is represented in perspective, using the figures to reduce the points of friction. In altering the perspective structure and inducing ambiguity in the optical measure, Raffaello indirectly contests the homogeneity of space and shifts the reception of the work from a purely optical level to a tactile one, calling the mind and other senses.

**Funding:** This research received no external funding.

**Institutional Review Board Statement:** Not applicable.

**Informed Consent Statement:** Not applicable.

**Data Availability Statement:** Not applicable.

**Acknowledgments:** This article combines, updates, and furtherly develops two former studies published in the book of Marcello Fagiolo, *Raffaello invisibile. Lo spazio, l'arco di trionfo, la cupola* (Roma: De Luca, 2020). All of the diagrams, drawings, and photomontages were made by the author. The author would like to express his most sincere gratitude to Marcello Fagiolo for the generous and proactive support in this research and to thank Marta Salvatore for her generous suggestions.

**Conflicts of Interest:** The author declares no conflict of interest.

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
