# Peer review of "Perspective Studies on Perugino’s and Raffaello’s Painted Architecture"

_arts, 2018_

Round 1
Reviewer 1 Report
Overall, this is a fascinating and important topic clearly being addressed by a specialist with technical expertise architectural drawing. I hope to see more work in this area--and not just on Renaissance painting, also Ancient Roman and other areas--and I look forward to having students of architectural history read about the results.
The paper makes ample reference to important and appropriate scholars in the field of Renaissance art such as White and Kemp. There are some more recent specific studies of fictive architecture, however that might bear of what is being done here even though they are not about Raphael and Perugino--for example, Zaru, "Creating a Devotional Space. Architectural Metaphors in Venetian Renaissance Altarpieces" (Artibus et Historiae, 2018), Lupi, "The Rhetoric of Fictive Architecture" (Architectural History, 2017), or the considerable literature on the so-called "Ideal Cities" in Baltimore, Berlin, and Urbino or even Masaccio's Trinity. There I suggest doing more thorough research in the area of fictive representations of architecture in the Italian Renaissance before publishing.
I further suggest that doing such research may permit conclusions that deeply impact the interpretion of the paintings under consideration as well as comparisons with the real architecture of the period, both of which are currently lacking. Such conclusions would be wecomed by the scholarly community.
Author Response
- In general, I mostly modified the Introductionto include the interesting suggestions of the referees and to frame better the procedure of perspective restitution. In this sense, I added a new figure 1 and renumbered the others. Other parts have been integrated, too and a dozen new books added to the Bibliography
REFEREE 1
Overall, this is a fascinating and important topic clearly being addressed by a specialist with technical expertise architectural drawing. I hope to see more work in this area--and not just on Renaissance painting, also Ancient Roman and other areas--and I look forward to having students of architectural history read about the results.
The paper makes ample reference to important and appropriate scholars in the field of Renaissance art such as White and Kemp. There are some more recent specific studies of fictive architecture, however that might bear of what is being done here even though they are not about Raphael and Perugino--for example, Zaru, "Creating a Devotional Space. Architectural Metaphors in Venetian Renaissance Altarpieces" (Artibus et Historiae, 2018), Lupi, "The Rhetoric of Fictive Architecture" (Architectural History, 2017), or the considerable literature on the so-called "Ideal Cities" in Baltimore, Berlin, and Urbino or even Masaccio's Trinity. There I suggest doing more thorough research in the area of fictive representations of architecture in the Italian Renaissance before publishing.
- The Introductionhas been enhanced to better frame the perspective painting and restitution in the historical context, which eventually included most of the texts kindly suggested by the referees.
- In particular, Mantegna’s work has been mentioned to demonstrate the effect of perspective application to existing figurative models. Lupi’s (like Davies’, also added to references) works are particularly addressed to semantic aspects, like the devotional role of perspective in painted architecture. Zaru’s is more concerning the paper, although focused on the illusory effects of perspective as a virtual expansion of real space, like Masaccio’s Trinity (whose interesting analysis by Camerota was quoted, too),
I further suggest that doing such research may permit conclusions that deeply impact the interpretion of the paintings under consideration as well as comparisons with the real architecture of the period, both of which are currently lacking. Such conclusions would be wecomed by the scholarly community.
- This paper presents only a small peek into wider research. I’m currently working on a book which is designed to expand the case studies in historical and geographical terms and to use the information from the perspective restitution for analysing designed and built architecture by Raphael and his many collaborators.

Reviewer 2 Report
I quite enjoyed the essay. it was easy to read and thoughtful.
In general I found this essay quite useful and the diagrams quite compelling as they provide rich material for consideration as we look at Early Modern art.
A few points to note.
1) the discussion is intimately tied to the idea of viewing perspective but in many instances the location of the viewer is not given importance (except in drawings and with Raffaello's Disputa. Take for example, the wall decorations of the Sistine and the predella images of the Fano Altarpiece. As we know to maintain continuity across the various artists that produced the frescoes of the Lives of Christ and Moses, they artists relied upon similar scales and visual construction (ie consistent color in the wardrobes of the protagonists across the frescoes and scenes for ease of legibility, the stacking of the foreground with main narrative, etc.) and the Perugino's image like those of his peers are elevated well above the floor. related we have Raphael's predella images adopting a uniform vanishing point. My question here, is how do the works account the viewer here? Does the perspective shift to account for an eye-level or elevated view engagement (thinking of your drawing for the Dispute). Some situating of the viewer would be relevant here.
2) Restitution. The word alone has specific connotations that dont necessarily conform to those in perspectival restitutions. I think defining the terms usage would be useful. I think for accessibility to larger audience and for clarity, might be worth have a reference to the term restitution architectural practices, or define it as found in Joanna Barbara Rapp’s article in The Journal of Architecture 13, no. 6 (2008): 701-736.
3) between lines 120-131 you speak of the pseudo basilica from the Coronation of Ana Silvio Piccolomini. To make the point stronger, it could be useful to mention the disproportioned framing devices that separates the various frescoes. The architectural elements there simply not rendered with same structural understanding as the basilica.
4) is it worth nothing that Raffaello worked on St. Peters as architect in 1515?
5) on several of your diagrams, it would be useful to have a), b), c) ... labeling for continuity and clarity across them particularly fig 9
6) likewise a bit longer description might be need in a few of the figure captions. for example fig. 10, what are the various viewing perspectives we are seeing? isometric, profile? This would provide great accessibility.
Author Response
- In general, I mostly modified the Introductionto include the interesting suggestions of the referees and to frame better the procedure of perspective restitution. In this sense, I added a new figure 1 and renumbered the others. Other parts have been integrated, too and a dozen new books added to the Bibliography
REFEREE 2
I quite enjoyed the essay. it was easy to read and thoughtful.
In general I found this essay quite useful and the diagrams quite compelling as they provide rich material for consideration as we look at Early Modern art.
A few points to note.
1) the discussion is intimately tied to the idea of viewing perspective but in many instances the location of the viewer is not given importance (except in drawings and with Raffaello's Disputa. Take for example, the wall decorations of the Sistine and the predella images of the Fano Altarpiece. As we know to maintain continuity across the various artists that produced the frescoes of the Lives of Christ and Moses, they artists relied upon similar scales and visual construction (ie consistent color in the wardrobes of the protagonists across the frescoes and scenes for ease of legibility, the stacking of the foreground with main narrative, etc.) and the Perugino's image like those of his peers are elevated well above the floor. related we have Raphael's predella images adopting a uniform vanishing point. My question here, is how do the works account the viewer here? Does the perspective shift to account for an eye-level or elevated view engagement (thinking of your drawing for the Dispute). Some situating of the viewer would be relevant here.
- For sure, the question of the position of the ideal point of view is generally relevant, especially when the perspective structure is evidenced by the architectural elements. For example, in the middle of XVI century, this will be the controversial issue of an interesting debate between the artists, in the context of the Milan Cathedral. Anyway, in the context of a single article, it is not possible to analyse all the cited works in detail. A single case, that of the Disputa, has been chosen to illustrate the number of elements that can influence the conception and perception of a complex work.
2) Restitution. The word alone has specific connotations that dont necessarily conform to those in perspectival restitutions. I think defining the terms usage would be useful. I think for accessibility to larger audience and for clarity, might be worth have a reference to the term restitution architectural practices, or define it as found in Joanna Barbara Rapp’s article in The Journal of Architecture 13, no. 6 (2008): 701-736.
- I quoted Rapp’s article, added a brief explanation of the procedure in the Introductionand a figure to illustrate a basic case.
3) between lines 120-131 you speak of the pseudo basilica from the Coronation of Ana Silvio Piccolomini. To make the point stronger, it could be useful to mention the disproportioned framing devices that separates the various frescoes. The architectural elements there simply not rendered with same structural understanding as the basilica.
- Thank you for this suggestion. I added a brief description of the fresco cycle to illustrate the role of the architectural frames in relationship with the scenes inside them.
4) is it worth nothing that Raffaello worked on St. Peters as architect in 1515?
- Actually, I forgot to mention this fundamental aspect. I added a pair of considerations on this. Thank you.
5) on several of your diagrams, it would be useful to have a), b), c) ... labeling for continuity and clarity across them particularly fig 9
- I inserted letters to make the identification easier.
6) likewise a bit longer description might be need in a few of the figure captions. for example fig. 10, what are the various viewing perspectives we are seeing? isometric, profile? This would provide great accessibility.
- I corrected and expanded some of the captions, in order to make them more uniform and expressive.

Round 2
Reviewer 1 Report
I think the revisions have substantially improved the paper.